# Beyond Circadian Patterns: Mechanistic Insights into Sleep–Epilepsy Interactions and Therapeutic Implications

**DOI:** 10.3390/cells14171331

**Published:** 2025-08-28

**Authors:** Kanghyun Kwon, Yoonsung Lee, Man S. Kim

**Affiliations:** 1Translational-Transdisciplinary Research Center, Clinical Research Institute, Kyung Hee University Hospital at Gangdong, College of Medicine, Kyung Hee University, Seoul 02447, Republic of Korea; kwonkh1015@khu.ac.kr; 2Department of Medicine, College of Medicine, Kyung Hee University, Seoul 02447, Republic of Korea

**Keywords:** epilepsy, sleep spindles, thalamocortical circuits, circadian rhythms, REM sleep

## Abstract

The relationship between sleep and epilepsy involves complex interactions between thalamocortical circuits, circadian mechanisms, and sleep architecture that fundamentally influence seizure susceptibility and cognitive outcomes. Epileptic activity disrupts essential sleep oscillations, particularly sleep spindles generated by thalamic circuits. Thalamic epileptic spikes actively compete with physiological sleep spindles, impairing memory consolidation and contributing to cognitive dysfunction in epileptic encephalopathy. This disruption explains why patients with epilepsy often experience learning difficulties despite adequate seizure control. Sleep stages show differential seizure susceptibility. REM sleep provides robust protection through enhanced GABAergic inhibition and motor neuron suppression, while non-REM sleep, particularly slow-wave sleep, increases seizure risk. These observations reveal fundamental mechanisms of seizure control within normal brain physiology. Circadian clock genes (*BMAL1, CLOCK, PER, CRY*) play crucial roles in seizure modulation. Dysregulation of these molecular timekeepers creates permissive conditions for seizure generation while being simultaneously disrupted by epileptic activity, establishing a bidirectional relationship. These mechanistic insights are driving chronobiological therapeutic approaches, including precisely timed antiseizure medications, sleep optimization strategies, and orexin/hypocretin system interventions. This understanding enables a paradigm shift from simple seizure suppression toward targeted restoration of physiological brain rhythms, promising transformative epilepsy management through sleep-informed precision medicine.

## 1. Introduction: The Bidirectional Sleep–Epilepsy Nexus

In 1885, Sir William Gowers observed that “of all conditions that favor the occurrence of epileptic seizures, sleep stands first.” More than a century later, this prescient observation has evolved into a sophisticated understanding of the bidirectional mechanisms linking sleep and epilepsy. Modern research has revealed that this relationship transcends simple temporal associations, involving complex interactions between sleep architecture, thalamocortical oscillations, and molecular timekeeping systems that fundamentally influence seizure generation, propagation, and cognitive consequences.

The clinical significance of sleep–epilepsy interactions is profound. Patients with epilepsy who report sleep disturbances exhibit a lower quality of life, with depression and anxiety as prevalent comorbidities [1]. Furthermore, sleep deprivation is a potent seizure trigger in patients with epilepsy, creating a vicious cycle in which poor sleep increases seizure risk, which in turn further deteriorates sleep quality [2]. This bidirectional relationship creates therapeutic challenges, while simultaneously offering novel interventional opportunities.

Recent technological advances in high-density electroencephalography, simultaneous thalamic–cortical recordings, and molecular profiling have revealed mechanistic details that are otherwise inaccessible to clinical investigations. These discoveries revealed that sleep and epilepsy share fundamental neural circuits, particularly thalamocortical networks, that generate both physiological sleep oscillations and pathological epileptic discharges [3,4,5]. Understanding these shared mechanisms provides unprecedented opportunities to develop targeted therapies that address both seizure control and sleep-related cognitive dysfunction.

This mechanistic focus represents a paradigm shift from descriptive to interventional approaches. However, much of this work remains siloed, with therapeutic developments still overwhelmingly focused on seizure suppression at the expense of addressing the profound cognitive and quality-of-life deficits linked to disrupted sleep physiology. This review directly confronts this gap, arguing that the next significant advance in epilepsy care will not come from a better spike suppressor, but from a strategy of actively restoring physiological sleep rhythms (spindles and coherent sleep architecture). We synthesize evidence from circuit neurophysiology, molecular biology, and clinical neurology to build a case in which a focus on restoring neurological health, rather than merely masking dysfunction, is both possible and essential.

## 2. Thalamocortical Networks: The Battleground Between Spindles and Spikes

### 2.1. Sleep Spindles as Guardians of Memory Consolidation

Thalamocortical circuits, comprising connections between thalamic nuclei and cortical regions, represent the fundamental neural networks that generate both physiological sleep oscillations and pathological epileptic discharges [3,4,5]. Sleep spindles, characteristic oscillations during non-REM sleep, represent one of the brain’s most precisely orchestrated rhythms [6]. Generated by GABAergic neurons in the thalamic reticular nucleus (TRN) and propagated through thalamocortical circuits, these brief bursts of synchronized activity serve as gatekeepers for memory consolidation and cortical plasticity [7]. The functional significance of sleep spindles extends beyond sleep maintenance to critical cognitive processes, with spindle density strongly correlating with learning capacity and memory retention in healthy individuals [8].

Sleep spindles work in coordination with K-complexes, another fundamental component of NREM sleep architecture. K-complexes are high-voltage, biphasic waveforms that represent isolated cortical downstates and serve dual functions as sleep-protective mechanisms and facilitators of memory consolidation [9,10]. Like sleep spindles, K-complexes are generated through thalamocortical interactions, with recent computational models demonstrating that K-complex generation involves disruption of thalamic spindling through corticothalamic feedback loops [9]. This intricate relationship between sleep spindles and K-complexes within the same thalamocortical circuits highlights the complexity of NREM sleep oscillations and their coordinated roles in sleep maintenance and cognitive function.

The bidirectional nature of thalamocortical circuits is fundamental to understanding both normal spindle generation and epileptic dysfunction. Corticothalamic projections, primarily originating from layer VI pyramidal neurons, provide essential feedback that modulates thalamic oscillations and maintains the coherent spindle activity necessary for memory consolidation [11,12]. This corticothalamic feedback creates a reverberating loop where cortical activity continuously shapes thalamic rhythms, while thalamic output drives cortical responses [13,14]. The strength and timing of this corticothalamic communication determine whether the network generates protective sleep spindles or pathological epileptic discharges, highlighting the critical importance of preserving these feedback mechanisms in therapeutic interventions.

Recent meta-analytical evidence has demonstrated that sleep spindles consolidate declarative memory through sophisticated tagging mechanisms that determine which information receives preferential processing during sleep [15,16]. The temporal clustering of spindles in “trains” during stage 2 non-REM sleep creates optimal neural conditions for memory reactivation, with the oscillatory quality of spindles directly linking brain state regulation to memory consolidation function [17,18]. Fast spindles (12–16 Hz) show particularly strong associations with procedural learning and processing speed, whereas their density correlates with sleep-dependent memory improvements [19,20].

The neurobiological machinery underlying spindle generation involves intricate interactions among three key components: thalamocortical relay neurons, TRN neurons, and cortical pyramidal cells. During the transition from wakefulness to non-REM sleep, reduced cholinergic input allows TRN neurons to enter burst-firing mode, generating rhythmic inhibitory postsynaptic potentials in thalamocortical neurons. This rhythmic inhibition, combined with the intrinsic membrane properties of thalamocortical cells, produces characteristic spindle oscillations that propagate to the cortex and are amplified by corticothalamic feedback loops [11,12].

Recent evidence has demonstrated that sleep spindles actively coordinate memory consolidation through precise temporal coupling with other sleep oscillations. Spindles provide temporal windows during which hippocampal sharp wave ripples can effectively transfer information to neocortical networks, facilitating the transformation of temporary hippocampal memories into stable cortical representations [21,22]. This coordination is crucial for procedural learning and declarative memory formation, which are frequently impaired in patients with epilepsy.

### 2.2. Epileptic Spikes as Circuit Hijackers

The mechanistic relationship between sleep spindles and absence seizures illustrates the fundamental vulnerability of corticothalamic circuits in epilepsy. Absence seizures, characterized by generalized 3-Hz spike-wave discharges, represent a pathological hijacking of the same corticothalamic networks that generate 12–16 Hz sleep spindles during normal non-REM sleep [23,24]. Both phenomena rely on synchronized oscillations between thalamic relay neurons, TRN cells, and cortical pyramidal neurons, but differ critically in their frequency characteristics and functional outcomes [25]. While spindles facilitate memory consolidation through precisely timed corticothalamic reverberations, absence seizures disrupt consciousness by overwhelming these same circuits with hypersynchronous activity [5]. This shared circuitry explains why children with absence epilepsy frequently exhibit altered sleep spindle characteristics and why corticothalamic dysfunction represents a common pathway in both sleep disorders and epileptic encephalopathies [26,27]. Groundbreaking research using simultaneous human thalamic and cortical recordings revealed that epileptic spikes actively disrupt sleep spindle generation via direct circuit interference [3]. Here, we define this as the core of thalamocortical dysfunction in epilepsy, a competitive process in which the circuits responsible for generating physiological and procognitive sleep rhythms are hijacked to produce pathological and proepileptic discharges (Figure 1). In patients with epileptic encephalopathy associated with spike wave activation during sleep (EE-SWAS), a condition characterized by cognitive dysfunction and developmental delay caused by frequent epileptic activity, thalamic epileptic spikes demonstrate a competitive relationship with physiological spindles; each spike significantly reduces the probability of subsequent spindle generation.

Competition occurs through multiple mechanisms. Epileptic spikes may induce excessive depolarization of thalamocortical neurons, preventing the hyperpolarization necessary for spindle initiation. Alternatively, spikes may act as “de facto spindles,” utilizing the same circuits and thereby prolonging the spindle-refractory period [28]. The clinical consequences are profound. Reduced spindle density correlates directly with the severity of cognitive impairment. This is not merely electrographic curiosity; it is a likely physiological substrate for the debilitating memory complaints and cognitive fog that haunt patients with epileptic encephalopathies [4,29].

Focal epilepsies demonstrate regionally specific spindle disruptions that precisely map seizure foci. In childhood epilepsy with centrotemporal spikes (CECTS), sleep spindle deficits are most pronounced in the centrotemporal regions during the active phase of the disease [4]. Remarkably, the spindle rate, but not the spike rate, predicts cognitive performance, suggesting that preserving physiological oscillations may be more therapeutically relevant than simply suppressing pathological activity. Recent studies in children with Rolandic epilepsy have confirmed that impaired sleep-dependent memory consolidation correlates specifically with reduced spindle density in affected brain regions [13].

Adult epilepsy populations show similar patterns; patients with focal epilepsy demonstrate reduced global and local spindle rates, which correlate with attention deficits [30]. The asymmetry of sleep spindle characteristics between hemispheres can serve as a lateralization tool for epileptic foci, with significant differences in spindle density and amplitude corresponding to seizure localization [14]. These findings establish sleep spindles as biomarkers of thalamocortical dysfunction and as potential therapeutic targets.

### 2.3. TRN: The Critical Hub

The thalamic reticular nucleus (TRN) has emerged as a critical nexus in the sleep–epilepsy relationship. As pacemakers for sleep spindles, this thin shell of GABAergic neurons that surrounds the dorsal thalamus is the chief regulator of physiological thalamocortical rhythms [31,32]. However, their pivotal position makes them points of profound vulnerability. Computational models have revealed that even subtle shifts in TRN connectivity or excitability can tip the network balance away from generating protective sleep spindles and toward producing pathological spike-wave discharges [33]. This suggests that the TRN is not merely a passive structure, but a highly sensitive switch point where the brain’s rhythmic activity can be steered toward either cognitive function or epileptic dysfunction.

The central role of the TRN is not governed by isolation. Its excitability is under the profound modulatory influence of both sleep-state neurochemistry and the master molecular clock [34,35]. The shift in cholinergic tone during REM sleep, for instance, directly alters TRN firing patterns, contributing to seizure suppression. Furthermore, as will be discussed, core clock genes regulate the very ion channels that determine TRN function, linking the brain’s 24-h cycle directly to this gatekeeper of thalamocortical activity. Therefore, therapeutic targeting of TRN holds promise not only for seizure control, but also for restoring the physiological sleep oscillations essential for cognition.

## 3. Sleep State-Dependent Seizure Susceptibility: The REM Sleep Paradox

Rapid eye movement (REM) sleep, characterized by high-frequency desynchronized cortical activity, vivid dreaming, and muscle atonia, represents a fascinating paradox in epilepsy. While the state of high-frequency desynchronized cortical activity resembles that of wakefulness, a period of relative seizure vulnerability, seizure frequency drops dramatically during REM sleep [36]. This apparent contradiction is resolved by unique neurochemical and physiological states that actively suppress seizure generation and propagation. Understanding the mechanisms underlying this powerful and naturally occurring anticonvulsant state offers a unique window into fundamental seizure control strategies that could inspire novel therapeutic developments. This section discusses components of the protective shield against REM sleep, from its distinct neurotransmitter profile to its profound influence on GABAergic tone. The distinct neurophysiological characteristics of each sleep state that contribute to these dramatic shifts in seizure susceptibility are outlined in Table 1.

### 3.1. Protective Power of REM Sleep

The neurochemical basis of REM sleep protection involves profound changes in neurotransmitter systems. During REM sleep, noradrenergic, serotonergic, and histaminergic neurons in the brainstem are virtually silent, whereas cholinergic activity in the pons and basal forebrain increases dramatically [37]. This neurotransmitter profile enhances GABAergic inhibition throughout the brain, raising the seizure threshold significantly above the levels observed during wakefulness or non-REM sleep [39]. However, this protective shield was not absolute. Some intracranial EEG studies have revealed that focal interictal activity, particularly within the mesiotemporal structures, can persist or even be paradoxically activated during REM sleep, suggesting that suppressive mechanisms may be less effective in chronically established epileptic networks.

Physiological studies using selective REM sleep manipulation have identified specific components responsible for seizure protection. Research has demonstrated that both characteristic muscle atonia (sleep paralysis) and thalamocortical EEG desynchronization of REM sleep contribute independently of seizure suppression [38]. Loss of muscle atonia eliminates protection against motor seizures, while preserving protection against generalized EEG seizures, whereas loss of EEG desynchronization has the opposite effect. This finding suggests that REM sleep operates through parallel protective mechanisms that target different aspects of seizure expression.

### 3.2. GABA-Mediated Inhibition During REM Sleep

The enhanced GABAergic tone characteristic of REM sleep represents a fundamental mechanism for seizure protection. During this sleep stage, both phasic and tonic GABA-mediated inhibition increase significantly, creating a hyperpolarized neuronal environment that opposes the synchronous depolarization necessary for seizure initiation [40]. This enhanced inhibition operates through multiple GABA receptor subtypes, with extrasynaptic δ-subunit-containing GABA-A receptors playing a particularly important role in mediating tonic inhibition [41].

Recent pharmacological studies have demonstrated that neurosteroids, which potently activate both synaptic and extrasynaptic GABA-A receptors, maintain their antiseizure efficacy during prolonged seizure activity when benzodiazepines lose their effectiveness [42,43]. This differential pharmacological sensitivity suggests that REM sleep seizure protection may involve neurosteroid-like mechanisms that maintain efficacy at membrane-bound GABA-A receptors, unlike benzodiazepines whose effectiveness is reduced by receptor internalization during status epilepticus.

Therefore, understanding REM sleep protection has significant clinical implications. Patients with REM sleep behavioral disorders who lose muscle atonia during REM sleep show altered seizure patterns that provide natural experiments on the importance of REM sleep mechanisms [44]. Additionally, medications that suppress REM sleep may inadvertently increase seizure susceptibility, suggesting that therapeutic approaches should consider REM sleep preservation as a clinical goal.

### 3.3. Sleep Architecture Disruption in Epilepsy

Epilepsy profoundly alters sleep architecture, with patients showing consistent reductions in REM sleep duration and efficiency. Meta-analyses have revealed that REM sleep reduction occurs across epilepsy syndromes, suggesting a fundamental disruption to sleep regulatory mechanisms rather than syndrome-specific effects [45]. REM sleep reduction may contribute to the cognitive impairments observed in epilepsy, as REM sleep plays a crucial role in emotional regulation, memory consolidation, and creative problem solving.

Drug-naïve patients with focal epilepsy exhibit decreased REM sleep and sleep efficiency compared with those with generalized epilepsy, who exhibit increased slow-wave sleep but similarly reduced sleep efficiency [46]. Recent findings demonstrate that REM sleep effects on interictal epileptic discharges vary significantly by brain region, with neocortical areas showing greater seizure suppression during REM sleep than mesiotemporal regions [47]. These architectural changes create a clinical phenotype in which a reduced REM sleep percentage serves as a potential biomarker for epilepsy diagnosis, with predictive models based on REM sleep metrics showing good discriminatory performance [48].

The temporal relationship between seizures and changes in sleep architecture provides insight into causation. Nocturnal seizures specifically disrupt subsequent REM sleep periods, whereas daytime seizures have less dramatic effects on subsequent sleep architecture [49]. This suggests that seizures occurring during vulnerable sleep stages have more profound effects on sleep regulation than those occurring during wakefulness, which potentially explains why patients with nocturnal seizures often report more severe sleep disturbances.

Antiseizure medications add another layer of complexity to sleep architecture disruption. Many traditional antiseizure drugs further suppress REM sleep, potentially exacerbating the sleep disturbances caused by epilepsy itself [50]. This creates a therapeutic dilemma in which seizure control may come at the cost of sleep quality, potentially contributing to the cognitive and psychiatric comorbidities observed in patients with epilepsy.

## 4. Molecular Clock Mechanisms in Epilepsy

### 4.1. Circadian Clock Genes as Seizure Modulators

The molecular circadian clock system consists of interlocking transcriptional–translational feedback loops involving core clock genes, including *BMAL1*, *CLOCK*, *PER1/2/3*, and *CRY1/2*, which generate approximately 24-h oscillations in gene expression that regulate numerous physiological processes relevant to seizure generation [51]. This system exerts a profound influence on seizure susceptibility and epileptic activity patterns. Recent research has revealed that disruption of these molecular timekeeping mechanisms increases seizure susceptibility and is disrupted by epileptic activity, creating complex bidirectional interactions. The specific roles of these core clock genes and their dysregulation in the context of epilepsy are summarized in Table 2.

*BMAL1* (Brain and Muscle ARNT-Like 1) emerges as a crucial player in the pathophysiology of epilepsy. In experimental temporal lobe epilepsy, *BMAL1* expression decreases significantly during both the latent and chronic phases, particularly in the hippocampal CA1 and dentate gyrus regions [52]. Selective knockout of *BMAL1* in hippocampal neurons lowers the seizure threshold and increases the susceptibility to chemically induced seizures, demonstrating a direct protective role for this core clock gene.

The *PER* gene family (*PER*1, *PER*2, and *PER*3) exhibits complex alterations in epilepsy models. *PER*2 expression, which normally oscillates with a robust circadian rhythm, becomes dysregulated following status epilepticus with altered phase relationships and reduced amplitude [54]. Importantly, *PER*2-knockout mice demonstrate increased seizure susceptibility, whereas *PER*2 overexpression can be protective, suggesting that maintaining proper *PER*2 function may represent a therapeutic target.

### 4.2. Circadian Control of Neuronal Excitability

The molecular clock machinery directly regulates multiple aspects of neuronal excitability through transcriptional control of ion channels, neurotransmitter receptors, and metabolic enzymes (Figure 2). Approximately 43% of protein-coding genes exhibit circadian expression patterns, including many genes crucial for neuronal function [55]. This extensive circadian regulation explains why seizure susceptibility varies dramatically across the 24-h cycle, even in the absence of obvious environmental cues.

The molecular clock robustly regulates voltage-gated ion channels, but its true impact comes from orchestrating a coordinated 24-h symphony of neuronal excitability. It does not merely turn single genes on and off; it also ensures that the channels promoting depolarization (e.g., sodium channels) and those promoting repolarization (e.g., specific potassium channels) have expression peaks offset from one another [56]. This temporal orchestration indicates that a neuron’s intrinsic excitability is not static; it systematically rises and falls across the day–night cycle, creating predictable windows of seizure vulnerability. This is exemplified by genes such as KCND2, whose mutation contributes to epilepsy and whose own expression is commanded by the core clock gene *PER*2. The brain’s primary inhibitory system is similarly governed by circadian oscillations in GABA-A receptor subunits, ensuring that the strength of inhibition also waxes and wanes [57,58]. This results in a brain with a constant seizure threshold and clock-controlled flux. While this framework is compelling, a crucial next step in the field is to demonstrate precisely how the disruption of a core clock gene, such as *BMAL1* directly alters the excitability and rhythmic firing of TRN neurons, thereby providing a definitive molecular-to-circuit mechanism for the spindle deficits observed in epilepsy.

### 4.3. Metabolic Rhythms and Seizure Susceptibility

The molecular clock coordinates brain metabolism through the direct transcriptional control of glycolytic enzymes, mitochondrial proteins, and metabolic sensors. This metabolic regulation creates time-of-day variations in ATP availability, glucose utilization, and oxidative stress resistance, which influence seizure thresholds [59]. During the normal resting phase, enhanced metabolic efficiency and reduced oxidative stress may contribute to decreased susceptibility to seizures.

Brain-derived neurotrophic factor (BDNF), a crucial regulator of neuronal survival and plasticity, exhibits a robust circadian regulation that is disrupted during epilepsy. Circadian oscillation of BDNF is controlled by *CLOCK*-*BMAL1* binding to E-box elements in the BDNF promoter [53]. In epilepsy models, circadian BDNF regulation is lost, potentially contributing to the neuronal hyperexcitability and impaired plasticity observed in epilepsy.

The relationship between circadian metabolism and antiseizure drug effectiveness represents an emerging area of clinical interest. Many antiseizure medications undergo circadian-dependent metabolism via hepatic cytochrome P450 enzymes, which show robust time-of-day expression patterns [60]. This suggests that optimal drug timing may enhance therapeutic efficacy while minimizing side effects, supporting the development of chronopharmacological approaches for the treatment of epilepsy.

## 5. Clinical Syndrome Spotlight: Paradigmatic Disorders

The mechanisms described above are not theoretical constructs, and their clinical importance is clearly illustrated by paradigmatic sleep-related epilepsy syndromes. Examining disorders such as Sleep-Related Hypermotor Epilepsy (SHE) and Juvenile Myoclonic Epilepsy provides direct evidence of how disrupted circuits and sleep–wake transitions drive seizure generation and its consequences. The defining clinical and sleep-related characteristics of these disorders, which serve as clear illustrations of the sleep–epilepsy nexus, are detailed in Table 3.

### 5.1. SHE: A Model Disorder

SHE is a paradigmatic disorder that illustrates an intimate connection between sleep architecture and seizure generation. Previously known as nocturnal frontal lobe epilepsy, SHE is characterized by complex hypermotor seizures arising exclusively from non-REM sleep, thus providing a natural model for understanding sleep-dependent seizure mechanisms [61]. Seizures typically emerge from stage 2 non-REM sleep, particularly when sleep spindles are most prominent, suggesting disruption to normal thalamocortical oscillations.

The genetic basis of autosomal dominant SHE involves mutations in nicotinic acetylcholine receptor subunits (CHRNA4, CHRNB2, and CHRNA2), which are highly expressed in thalamocortical circuits [62]. These mutations alter the balance between excitation and inhibition during sleep, creating a pathophysiological environment in which normal sleep oscillations are transformed into epileptic discharges. The crucial role of the cholinergic system in the sleep–wake transition and REM sleep generation provides mechanistic insights into why these mutations specifically affect sleep-related seizures.

Recent therapeutic approaches for SHE have improved our understanding of its underlying pathophysiology. Nicotine, acting as a partial agonist of the affected receptor subunits, can paradoxically reduce seizure frequency in some patients with genetic SHE [63]. This counterintuitive finding demonstrates that a mechanistic understanding can lead to novel therapeutic strategies that may not emerge from conventional screening approaches.

### 5.2. Idiopathic Generalized Epilepsy and Sleep Disruption

Idiopathic generalized epilepsy demonstrates consistent alterations in sleep architecture that contribute to seizure susceptibility and cognitive dysfunction. Meta-analysis studies have revealed reduced sleep efficiency, prolonged sleep latency, and altered sleep stage proportions in patients with idiopathic generalized epilepsy [27]. These changes appear early in the disease course and persist despite adequate seizure control, suggesting fundamental alterations involving sleep regulatory mechanisms.

Juvenile myoclonic epilepsy, the most common idiopathic generalized epilepsy syndrome, is characterized by profound sleep architecture abnormalities. Such patients demonstrate reduced slow-wave sleep, fragmented sleep continuity, and altered sleep spindle characteristics [26]. Typical morning seizure clustering in this syndrome correlates with specific sleep-stage transitions, particularly the transition from sleep to wakefulness, when cholinergic arousal systems overcome sleep-promoting mechanisms.

The relationship between photosensitivity and sleep deprivation in idiopathic generalized epilepsy provides insights into the underlying mechanisms. Sleep deprivation enhances photoparoxysmal responses and lowers seizure thresholds, suggesting that normal sleep protects against excessive sensory-driven cortical synchronization [64]. This relationship supports therapeutic approaches that prioritize sleep hygiene as a fundamental component of seizure management.

## 6. Therapeutic Innovations: Chronobiological Approaches

### 6.1. Precision Timing of Antiseizure Medications

The therapeutic approaches described in this section represent emerging strategies at various stages of development, from preclinical investigation to early clinical trials. While these chronobiological interventions show promise, most require substantial additional research to establish safety, efficacy, and optimal implementation protocols before widespread clinical adoption. Recognition that seizure susceptibility varies predictably across circadian cycles has prompted the development of chronotherapeutic approaches—therapeutic strategies that optimize treatment timing based on biological rhythms—for antiseizure medication delivery [65]. Traditional antiseizure drug regimens typically maintain constant plasma levels through multiple daily doses; however, emerging evidence suggests that matching drug delivery to individual seizure patterns may potentially enhance efficacy while reducing side effects.

Chronopharmacological studies have revealed that antiseizure drug effectiveness varies significantly with the timing of administration [66,67]. Carbamazepine has enhanced bioavailability when administered in the evening, whereas phenytoin demonstrates superior seizure control in the morning [65,66]. These differences likely reflect circadian variations in drug absorption, distribution, metabolism, and target receptor sensitivity, all of which influence therapeutic outcomes. Recent clinical studies have demonstrated that the optimization of medication timing based on circadian rhythms can improve conversion rates and patient survival, particularly in drug-resistant epilepsy [68,69].

Advanced drug delivery systems now enable the precise temporal control of medication release. Modified-release formulations can be potentially designed to deliver higher drug concentrations during periods of increased seizure risk, while maintaining lower levels during protective sleep phases [70,71]. This approach requires careful characterization of individual seizure patterns using long-term monitoring; however, preliminary studies have suggested possible improvements in seizure control with a reduced medication burden, though larger clinical trials are needed to establish efficacy. Circadian-based dosing strategies show particular promise for seizure clusters, with diazepam nasal spray demonstrating clear circadian periodicity in efficacy [72].

### 6.2. Sleep Optimization Strategies

Therapeutic interventions targeting sleep quality and architecture represent a fundamental but underutilized approach to seizure management [73,74,75]. Sleep hygiene education, cognitive behavioral therapy for insomnia (CBT-I), and targeted pharmacological sleep enhancement may contribute to improved seizure control. The bidirectional nature of sleep–epilepsy interactions suggests that optimizing sleep may break the vicious cycle of poor sleep, promoting seizures, and seizures that disrupt sleep.

Clinical studies have demonstrated significant sleep disturbances across epilepsy populations, with subjective sleep complaints affecting up to 90% of patients and objective polysomnographic abnormalities present even in well-controlled epilepsy patients [76,77,78]. Sleep efficiency, total sleep time, and sleep architecture show consistent alterations correlated with seizure frequency and cognitive dysfunction [48,79,80]. These findings support sleep optimization as a primary therapeutic target rather than merely addressing sleep as a comorbidity.

Specific sleep interventions are promising for different populations with epilepsy. Continuous positive airway pressure (CPAP) therapy for obstructive sleep apnea in patients with epilepsy can significantly reduce seizure frequency, likely through multiple mechanisms, including improved sleep continuity and reduced sleep fragmentation [78,81]. Similarly, the treatment of restless legs syndrome and periodic limb movements can improve both sleep quality and seizure control.

Pharmacological sleep enhancement requires careful consideration of antiseizure drug interactions and seizure threshold effects [82,83]. Medications that enhance slow-wave sleep, such as sodium oxybate, show preliminary promise in potentially improving both sleep quality and seizure control in selected patients. However, REM sleep-suppressing medications may paradoxically increase seizure risk, highlighting the importance of understanding sleep stage-specific effects. The reciprocal relationship between sleep quality and seizure control necessitates comprehensive sleep assessment and targeted intervention in all patients [84].

### 6.3. Novel Molecular Targets

Recent mechanistic insights have identified the orexin/hypocretin system, a key regulator of sleep–wake transition and arousal, as a promising therapeutic target [85]. Given its central role, interventions targeting this system could theoretically offer the dual benefit of stabilizing sleep and reducing seizure susceptibility [86,87]. This prospect is supported by intriguing, albeit preclinical, evidence. In some genetic epilepsy models, orexin agonists have been shown to suppress spike-wave discharges, while orexin receptor antagonists can improve sleep quality and decrease seizure frequency [85,88]. Clinically, the disruption of orexin levels following status epilepticus further suggests the involvement of this system in human epilepsy [88,89].

However, translating these findings to clinical settings requires caution. Although orexin antagonists have been approved for insomnia, their application in diverse epilepsy populations presents challenges [87]. The effect of broadly suppressing arousal via orexin antagonism on daytime cognitive function, mood, and seizure control in patients with complex, chronic epilepsy remains a critical, unaddressed question. The preclinical data are complex, with both agonists and antagonists demonstrating anti-seizure effects depending on the model [85,88]. This highlights our incomplete understanding and underscores the fact that the orexin system is not a simple “on/off” switch for seizures. Its promise lies in its potential for targeted modulation; however, significant work is needed to define which patient populations might benefit from and mitigate the potential adverse effects of altering a fundamental regulator of vigilance [90,91].

Melatonin and melatonin receptor agonists represent another chronobiological intervention with sleep-promoting and antiseizure properties. Multiple clinical studies have demonstrated that melatonin supplementation can improve sleep quality and reduce seizure frequency in pediatric epilepsy populations [92,93,94,95,96,97,98,99,100,101]. The antioxidant and neuroprotective properties of melatonin may contribute to these benefits beyond simple sleep improvement, particularly in patients with refractory epilepsy. High-dose melatonin is particularly effective as an adjunct therapy for severe myoclonic epilepsy, with sustained benefits observed over extended treatment periods [102].

Modulation of sleep spindle generation represents a promising but largely experimental therapeutic approach. Positive allosteric modulators of extrasynaptic GABA-A receptors may potentially enhance sleep spindle amplitude and density while providing seizure protection [103,104,105,106,107]. This approach could theoretically simultaneously address the sleep spindle deficits observed in epileptic encephalopathies and provide anticonvulsant effects through enhanced tonic inhibition, though clinical validation remains limited.

### 6.4. Biomarker-Guided Interventions: The Immense Gap Between Data and the Clinic

The ultimate goal of this research is personalized medicine. However, the path from biomarker discovery to clinical utility is limited by practical and economic roadblocks that are often understated.

Sleep Spindles: While elegant measure of thalamocortical integrity [28], the use of spindles as clinical biomarkers is fundamentally a niche concept. It requires expert-led polysomnography, which is a costly and labor-intensive procedure that is unavailable to the vast majority of patients. Furthermore, the development of automated spindle-detection algorithms that are robust to artifacts and epileptiform activity endemic to clinical EEG remains a significant computational challenge. Without an inexpensive, scalable, and reliable method of detection, sleep spindles remain a powerful research tool but do not represent a common clinical tool.Circadian Biomarkers: Using wearables to guide chronotherapy is technologically feasible [108,109,110], but clinically and commercially stalled. The primary barrier is the lack of clear business cases for developing and validating these strategies. This requires pharmaceutical companies to fund complex and expensive trials to prove that the tailored dosing of their existing drugs is superior to standard regimens, a high-risk, low-reward proposition that few have pursued.Neurostimulation: Integrating chronobiological data into responsive neurostimulation devices is at the frontier of personalized therapy [111]. However, its relevance is restricted to a small fraction of the most refractory patients owing to the extreme cost and invasive nature of the therapy. The computational burden of analyzing continuous data streams from such devices is also immense, requiring a dedicated data science infrastructure. This will not be a scalable solution for epilepsy care in the near future.

## 7. Future Directions: Emerging Horizons

### 7.1. Personalized Chronotherapy

The future of epilepsy treatment lies in personalized chronotherapeutic approaches that consider individual circadian profiles, seizure patterns, and sleep architecture abnormalities. Advances in wearable technology and artificial intelligence have enabled the continuous monitoring of physiological parameters that inform real-time treatment adjustments [112]. Machine learning algorithms can identify subtle patterns in multidimensional data to predict seizure risk and optimize intervention timing.

Profiling of circadian clock genes will increasingly inform therapeutic decisions. Patients with specific clock gene variants may benefit from targeted chronobiological interventions, such as enhanced light therapy for those with delayed phase tendencies or modified melatonin receptor agonists for those with altered melatonin sensitivity [113]. This pharmacogenomic approach to chronotherapy represents the natural evolution of the principles of personalized medicine.

The integration of sleep monitoring into implantable devices opens possibilities for closed-loop therapeutic systems that adjust stimulation parameters based on sleep stage and circadian phase. Responsive neurostimulation systems can enhance sleep spindle generation during appropriate sleep stages, while providing seizure detection and intervention during vulnerable periods [114]. This integration of sleep physiology with neurostimulation represents a paradigm shift toward more naturalistic therapeutic approaches.

### 7.2. Sleep-Based Seizure Prediction

Emerging evidence suggests that changes in sleep architecture precede seizures in some patients, opening up possibilities for sleep-based seizure prediction [115]. Alterations in sleep spindle characteristics, slow-wave activity, and REM sleep patterns may serve as early warning signs of impending seizures and enable proactive therapeutic interventions.

Advanced signal processing techniques applied to sleep EEG data can extract subtle features that predict seizure risk hours to days in advance [116,117]. These approaches move beyond traditional seizure prediction based on interictal spike patterns toward a holistic assessment of brain state dynamics during sleep. The integration of multiple physiological signals such as EEG signals, heart rate variability, movement patterns, and autonomic measures may provide robust seizure prediction algorithms.

Clinical implementation of sleep-based seizure prediction requires the development of user-friendly monitoring systems and clear intervention protocols. Patients may receive warnings of increased seizure risk with recommendations for enhanced safety precautions, rescue medication use, or sleep schedule modifications [118]. This predictive approach could significantly improve quality of life while reducing seizure-related injuries and hospitalizations.

### 7.3. Gene Therapy: A Distant and Uncertain Horizon

While understanding the role of molecular clocks in epilepsy logically points toward gene therapy, this approach must be viewed with extreme caution as a distant and highly uncertain prospect. The concept of using viral vectors to correct *BMAL1* deficiency or CRISPR to edit clock gene variants is intellectually appealing but faces staggering translational barriers [119,120].

However, the primary obstacle is safety. The brain’s clock machinery is pleiotropic, indicating that these genes influence countless other critical cellular processes beyond excitability. Germline editing is ethically untenable and somatic editing in the adult brain carries an immense risk of off-target effects, oncogenesis from viral vectors, and irreversible disruption to essential neural functions. Furthermore, most epilepsies are not monogenic disorders and are thus not amenable to single-gene fixation. These are complex network diseases in which correcting one gene in one cell type may have little effect on the established pathology. While pharmacological “clock-stabilizing” drugs are a more feasible near-term goal, the development of central nervous system (CNS)-penetrant molecules with sufficient specificity remains a major challenge. At present, gene therapy for sleep–epilepsy disorders should be considered a conceptual goal that drives basic science and does not represent a realistic therapeutic strategy for the foreseeable future.

## 8. Conclusions: From Seizure Suppression to Rhythm Restoration

We reviewed the intricate dance between sleep and epilepsy, from the circuit-level competition between spindles and spikes to the molecular gears of the circadian clock. This evidence compels a departure from the decade-old treatment paradigm. The central argument of this review is that the prevailing focus on seizure suppression is incomplete and ultimately inadequate. It treats the most dramatic symptoms while ignoring the underlying collapse of physiological brain activity, which devastates cognition and quality of life. Epilepsy management cannot be treated with another anti-convulsant; it must be a procognitive, rhythm-restoring therapy (Figure 3).

The integration of these fields revealed a clear therapeutic target: the dysfunctional thalamocortical network. This is not an abstract concept; it is a measurable system in which health is reflected in the integrity of sleep spindles. Therefore, the immediate mandate for this field is twofold.

For clinicians, the call to action is to treat sleep dysfunction as a primary therapeutic target and not as a comorbidity. This requires routine, objective assessment of sleep architecture and prioritization of treatments—behavioral, pharmacological, or device-based—that protect or enhance REM sleep and non-REM stability. Choosing an antiseizure medication should no longer be based solely on seizure efficacy, but also on its impact on sleep architecture.

For researchers, the challenge is to move beyond preclinical models of seizure reduction and toward models of rhythm restoration. The primary endpoint in drug and device development should not be merely “fewer seizures,” but “improved spindle density,” “normalized REM sleep,” and “enhanced sleep-dependent memory consolidation.” This requires a new generation of clinical trials and the validation of sleep-based biomarkers as core outcome measures.

In 1885, Gowers identified sleep as a condition that “favors” seizures. More than a century later, this relationship was reframed. Healthy sleep does not favor epilepsy; it is the brain’s endogenous and powerful antiepileptic. The loss of this protection leads to disease. Therefore, the goal is not just to stop the seizures, but also to restore patients back to sleep.

## Figures and Tables

**Figure 1 cells-14-01331-f001:**
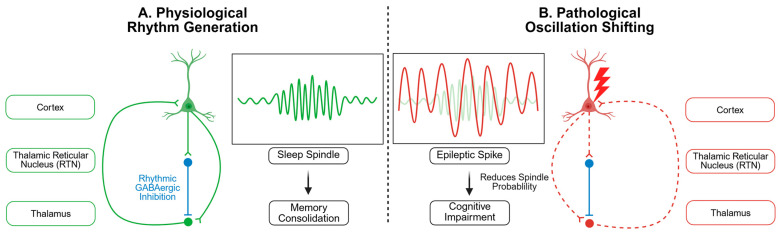
The Thalamocortical Circuit in Health and Disease. (**A**) During normal sleep, thalamocortical circuits generate spindles to support memory consolidation. (**B**) In epilepsy, this same circuit is hijacked by pathological spikes, which competitively suppress spindle generation and impair cognition.

**Figure 2 cells-14-01331-f002:**
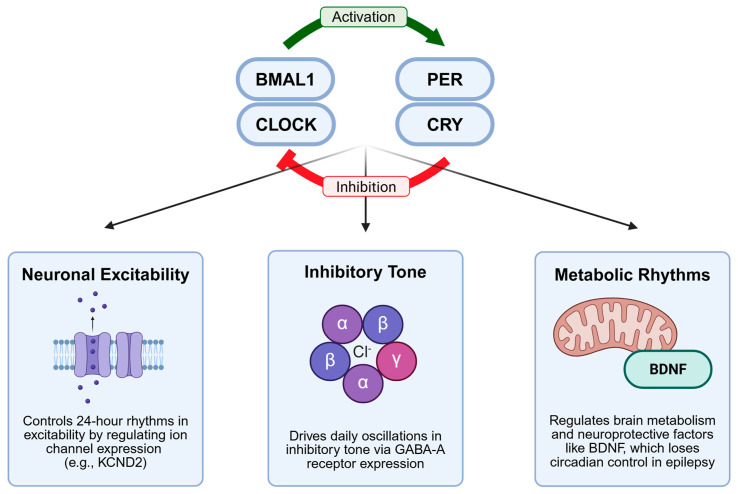
Circadian Regulation of Seizure Susceptibility. The core molecular clock (*BMAL1*, *CLOCK*, *PER*, *CRY*) governs seizure thresholds by driving 24-h rhythms in key downstream systems, including neuronal excitability, inhibitory tone, and brain metabolism.

**Figure 3 cells-14-01331-f003:**
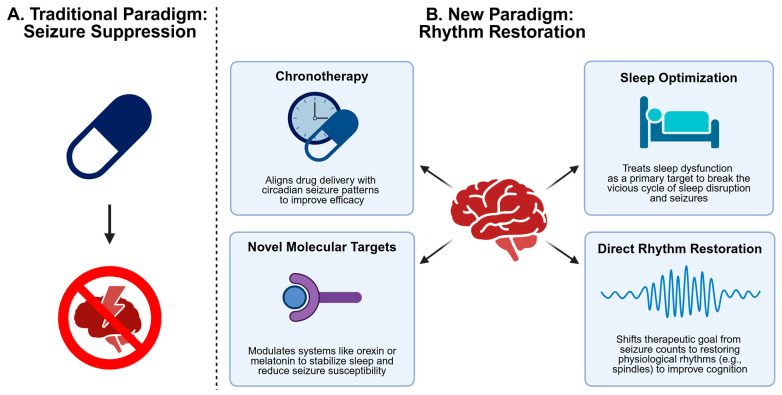
A Paradigm Shift in Epilepsy Management. (**A**) The traditional paradigm focuses narrowly on seizure suppression. (**B**) The new paradigm is a holistic strategy aimed at restoring physiological brain rhythms through chronotherapy, sleep optimization, novel molecular targets, and direct rhythm restoration.

**Table 1 cells-14-01331-t001:** Comparison of Sleep Stage Characteristics and Seizure Susceptibility.

Feature	Wakefulness	Non-REM Sleep	REM Sleep
Cortical EEG (Electroencephalography) Activity	Desynchronized, high frequency	Synchronized oscillations	Desynchronized, high-frequency (wakeful-like)
Dominant Neurotransmitters	High cholinergic and monoaminergic tone	Reduced cholinergic and monoaminergic activity	High cholinergic; noradrenergic and serotonergic neurons silent
Seizure Susceptibility	Relative seizure vulnerability	High; promotes seizure generation and propagation	Dramatically reduced; natural anticonvulsant state
Motor Activity	Normal muscle tone	Reduced muscle tone	Muscle atonia (sleep paralysis)
Reference	[37]	[36,37]	[38]

**Table 2 cells-14-01331-t002:** Role of Core Circadian Clock Genes in Epilepsy.

Gene	Normal Function in Circadian Rhythm	Dysregulation and Role in Epilepsy	Reference
*BMAL1*	Core positive regulator; heterodimerizes with *CLOCK* to activate target gene transcription	Decreased in temporal lobe epilepsy. Knockout lowers seizure threshold; gene has direct protective role	[51,52]
*CLOCK*	Core positive regulator; heterodimerizes with *BMAL1* to drive clock-controlled genes, including *BDNF*	Dysregulation disrupts *CLOCK-BMAL1* complex, causing loss of circadian BDNF regulation in epilepsy	[51,53]
*PER2*	Core negative regulator in transcriptional–translational feedback loop; inhibits *CLOCK/BMAL1* activity	Dysregulated following status epilepticus. Knockout increases seizure susceptibility; overexpression is protective	[51,54]
*CRY1/2*	Core negative regulators; inhibit *CLOCK/BMAL1* activity with *PER* proteins in feedback loop	Dysregulation creates permissive environment for seizure generation and disrupts epileptic activity patterns	[51]

**Table 3 cells-14-01331-t003:** Clinical and Sleep-related Features of Paradigmatic Epilepsy Syndromes.

Syndrome	Key Clinical Features	Primary Sleep-Related Characteristics	Proposed Underlying Mechanism
Sleep-related Hypermotor Epilepsy (SHE)	Complex hypermotor seizures arising exclusively from sleep	Seizures emerge from Stage N2 non-REM sleep. Timing coincides with periods when sleep spindles are most prominent	Genetic mutations in nicotinic acetylcholine receptor subunits
Juvenile Myoclonic Epilepsy (JME)	Most common idiopathic generalized epilepsy syndrome; characterized by morning seizure clustering	Patients exhibit profound sleep architecture abnormalities, including reduced slow-wave sleep and fragmented sleep continuity. Seizures correlate with sleep-to-wake transition. Sleep deprivation is a potent seizure trigger	Fundamental alterations in sleep regulatory mechanisms

## Data Availability

Not applicable.

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
