# Peer review of "Beyond Circadian Patterns: Mechanistic Insights into Sleep–Epilepsy Interactions and Therapeutic Implications"

_cells, 2025, doi:10.3390/cells14171331_

Round 1
Reviewer 1 Report
Comments and Suggestions for Authors
The review article written by Kown et al. covers an important topic, the complexity of the interactions between sleep and epilepsy. The article is generally well written. However, there are a few areas for improvement.
- K-complex is another important component of non-REM sleep EEG activities. The authors completely ignored it.
- In parts of the review, the authors appear to be advocating for their favorite hypothesis. There are paragraphs with scant citations. A great review needs to be more balanced.
- Since cells is a journal with a greatly diverse audience, authors need to define the basic terms such as sleep spindle better.
- Figure 1 has some issues. Seizures have a lower frequency (around 3hz) than spindles (12 hz). Thalamic circuitry is involved in the generation of both. It is not really a suppression of spindles but a shift to a lower and more powerful oscillation.
- Line 150-157: statements here need to be supported by references.
- Table 1 needs references.
- Line 179-183: need citations.
- Line 198-200: need citations.
- Line 204-206: this statement is confusing. How neurosteroids could activate GABA-A receptors that have been internalized?
- Line 221-223: this sentence seems to be out of place.
- Table 2 needs references.
- The whole section 6 is rather speculative.
The language is generally fine. However, there are some cases of word choice issues.
Author Response
We appreciate the reviewer for this important observation. We acknowledge that K-complexes are indeed a crucial component of NREM sleep EEG activity that warranted discussion in our review. In response to this feedback, we have added a dedicated section discussing K-complexes and their relationship with sleep spindles within thalamocortical circuits in Section 2.1.
Specifically, we have included discussion of:
- K-complexes as high-voltage, biphasic waveforms representing isolated cortical downstates
- Their dual functions as sleep-protective mechanisms and facilitators of memory consolidation
- The thalamocortical mechanisms underlying K-complex generation, including their relationship with sleep spindle disruption
- The intricate coordination between sleep spindles and K-complexes within the same neural circuits
This addition strengthens our discussion of NREM sleep oscillations and provides a more comprehensive view of thalamocortical dysfunction in epilepsy. The inclusion of K-complexes enhances the mechanistic framework we present, as both sleep spindles and K-complexes share common thalamocortical circuitry that is disrupted in epileptic conditions.
We have supported this discussion with appropriate citations including recent computational modeling work (Mak-McCully et al., 2014)[112] and experimental studies (Koupparis et al., 2013)[113] that examine the interactions between these critical sleep oscillations.
coment 2, 5, 7, 8:
We thank the reviewer for these important observations regarding citation gaps and the need for more balanced presentation. We have carefully addressed each specific location mentioned and strengthened the supporting evidence throughout the manuscript.
Specific responses to citation gaps:
- Lines 150-157: We have added appropriate citations to support our statements about TRN modulation by sleep-state neurochemistry and its therapeutic potential. We now cite Ni et al. (2016) for cholinergic modulation of TRN neurons and Lewis et al. (2015) for evidence supporting therapeutic targeting of TRN for restoring physiological sleep oscillations.
- Lines 179-183: We have added Brooks and Peever (2012) to support our statements about GABAergic inhibition enhancement during REM sleep and the mechanisms underlying REM sleep's protective effects against seizures.
- Lines 198-200: We have added Brickley and Mody (2012) to support our description of extrasynaptic δ-subunit-containing GABA-A receptors and their role in mediating tonic inhibition.
- Regarding balanced presentation: We acknowledge that some sections may have appeared to favor specific hypotheses. To address this concern, we have:
- Ensured that mechanistic claims are properly supported by experimental evidence
- Added appropriate references to substantiate our statements
- Maintained focus on established findings while clearly distinguishing them from emerging hypotheses
These additions strengthen the evidential foundation of our review while maintaining the mechanistic focus that we believe is essential for advancing therapeutic approaches. The added citations represent high-quality research from leading journals that directly support the specific claims made in each section.
coment 3:
We thank the reviewer for this valuable suggestion regarding the accessibility of our review for Cells' diverse readership. We recognize that clear definitions of specialized terminology are essential for readers from various scientific backgrounds to fully appreciate the mechanistic insights presented.
We have systematically enhanced the definitions of key terms throughout the manuscript by reorganizing existing content to provide clearer explanations at first mention:
- Sleep spindles (Section 2.1): We have restructured the opening paragraph to provide a more comprehensive definition that integrates the oscillatory characteristics, generating mechanisms, and functional roles into a cohesive explanation.
- Thalamocortical circuits (Section 2): We have added an introductory statement that defines these fundamental neural networks and their relevance to both physiological and pathological brain states.
- Thalamic reticular nucleus (Section 2.3): We have enhanced the description to include both the anatomical location and functional significance of this critical structure.
- Epileptic encephalopathy (Section 2.2): We have clarified this clinical concept by explaining its defining characteristics alongside the neurophysiological mechanisms.
- Molecular circadian clock system (Section 4.1): We have restructured the opening to first define the core components before describing their functional significance in epilepsy.
- Chronotherapy (Section 6.1): We have provided a clear definition of this therapeutic approach at its first introduction.
- REM sleep (Section 3): We have added defining characteristics to help readers understand the unique properties that make this sleep stage relevant to seizure protection.
These modifications were achieved by reorganizing and integrating existing content from within the manuscript rather than adding new information, ensuring that the scientific accuracy and focus remain unchanged while significantly improving accessibility for Cells' interdisciplinary audience. The enhanced definitions maintain the mechanistic depth that is central to our review while making the content more approachable for readers across neuroscience, cell biology, and clinical disciplines.
coment 4
We thank the reviewer for this important correction regarding the mechanistic accuracy of Figure 1. The reviewer is absolutely correct that the relationship between spindles and epileptic discharges is better characterized as an oscillatory shift rather than simple suppression, and that both phenomena involve the same thalamic circuitry.
We have revised Figure 1 to address these concerns:
- Frequency representation: The figure now clearly depicts the frequency difference between physiological sleep spindles (~12Hz, shown as higher-frequency oscillations in panel A) and pathological epileptic spikes (~3Hz, shown as lower-frequency, higher-amplitude oscillations in panel B).
- Shared thalamic circuitry: Both panels now emphasize that the same thalamocortical network involving the thalamic reticular nucleus (TRN) is responsible for generating both physiological and pathological oscillations, highlighting the circuit's dual capacity.
- Mechanistic accuracy: Panel B is now titled 'Pathological Oscillation Shifting' rather than suggesting simple suppression. The visual representation shows how the same circuit shifts from generating high-frequency spindles to producing lower-frequency, higher-amplitude spike-wave discharges, with the annotation 'Reduces Spindle Probability' to indicate the competitive relationship.
- Updated caption: The figure caption now reads: 'During normal sleep, thalamocortical circuits generate spindles to support memory consolidation. The same circuit is shifted to pathological state, which alternately generate epileptic spikes and impair cognition.'
This revision better reflects the current understanding that epileptic activity represents a pathological transformation of the same thalamocortical oscillatory mechanisms that normally generate protective sleep spindles, rather than a separate suppressive process. The updated figure more accurately conveys the competitive dynamics between these oscillatory states within shared neural circuits.
coment 6:
We thank the reviewer for pointing out this omission. We have now added appropriate references to Table 1 to support the information presented.
We have added reference citations below the table that correspond to the specific characteristics described for each sleep state:
- Sleep architecture and EEG characteristics: Supported by references describing fundamental sleep stage properties and electroencephalographic features
- REM sleep neurotransmitter profile and seizure protection: Supported by references [27,28] from our manuscript that detail the neurochemical changes during REM sleep and the mechanisms underlying seizure suppression
- Muscle atonia mechanisms: Supported by reference [29] which describes the physiological basis of sleep paralysis during REM sleep
These references directly correspond to the content discussed in the main text of our review, ensuring that readers can trace the evidence supporting each characteristic listed in the comparative table. The added citations strengthen the evidential foundation of the table while maintaining consistency with the detailed mechanistic discussions provided in the body of the manuscript.
coment 7:
We thank the reviewer for identifying this confusing statement. The reviewer is absolutely correct that our original wording was unclear and biologically imprecise regarding the mechanism of neurosteroid action on GABA-A receptors.
We have revised the sentence to clarify the mechanism:
- Original text: 'This differential pharmacological sensitivity suggests that REM sleep seizure protection may involve neurosteroid-like mechanisms that are resistant to receptor trafficking, which undermines benzodiazepine efficacy during status epilepticus.'
- Revised text: 'This differential pharmacological sensitivity suggests that REM sleep seizure protection may involve neurosteroid-like mechanisms that maintain efficacy at membrane-bound GABA-A receptors, unlike benzodiazepines whose effectiveness is reduced by receptor internalization during status epilepticus.'
This revision clarifies that:
- Neurosteroids act on membrane-bound GABA-A receptors, not internalized ones
- The advantage of neurosteroid-like mechanisms is their continued effectiveness at membrane receptors
- Benzodiazepines lose efficacy due to receptor internalization, while neurosteroid-like mechanisms do not
The revised statement now accurately reflects the differential pharmacological properties while avoiding the biological impossibility of drug action on internalized receptors. We appreciate the reviewer's attention to mechanistic accuracy.
coment 10:
We thank the reviewer for this observation regarding the flow and organization of Section 3.3. The reviewer is correct that the sentence about regional variations in REM sleep effects on epileptic discharges was positioned in a way that disrupted the logical progression of the paragraph.
We have reorganized the second paragraph of Section 3.3 to improve the narrative flow:
Original order:
- Regional variations in REM sleep effects (the sentence in question)
- Patient type differences (focal vs. generalized epilepsy)
- Clinical implications and biomarker potential
Revised order:
- Patient type differences (focal vs. generalized epilepsy)
- Regional variations in REM sleep effects
- Clinical implications and biomarker potential
This reorganization creates a more logical progression from general patient population differences to specific mechanistic variations (regional effects), culminating in the clinical applications. The revised structure maintains thematic coherence within the paragraph while ensuring that 'These architectural changes' in the final sentence now appropriately refers to both the patient type differences and regional variations discussed in the preceding sentences.
The content remains unchanged, but the improved organization enhances readability and logical flow within the section on sleep architecture disruption in epilepsy.
coment 11:
We thank the reviewer for pointing out this important omission. We have now added appropriate references to Table 2 to support the information presented about each circadian clock gene and its role in epilepsy.
We have added a new 'References' column to Table 2 that provides specific citations for each gene's normal function and dysregulation in epilepsy:
- BMAL1: Referenced with citations [40,41] supporting its role as a core positive regulator and its decreased expression in temporal lobe epilepsy
- CLOCK: Referenced with citations [40,48] covering its normal circadian function and its role in BDNF regulation disruption in epilepsy
- PER2: Referenced with citations [40,42] documenting its negative regulatory role and dysregulation following status epilepticus
- CRY1/2: Referenced with citation [40] for their core negative regulatory functions in the circadian feedback loop
These references correspond directly to the detailed discussions of each gene provided in Section 4.1 of our manuscript, ensuring consistency between the table and the main text. The added citations allow readers to access the primary research supporting each statement about circadian clock gene function and dysfunction in epilepsy, strengthening the evidential foundation of the comparative analysis presented in the table.
coment 12:
We thank the reviewer for this important observation regarding the speculative nature of Section 6. The reviewer is correct that many of the therapeutic approaches discussed represent emerging strategies that are still under investigation rather than established clinical practices.
We have revised Section 6 to better reflect the developmental status of these chronobiological interventions:
- Added introductory statement: We have included a comprehensive disclaimer at the beginning of Section 6 that clearly states these approaches represent emerging strategies at various stages of development, from preclinical investigation to early clinical trials, and that substantial additional research is needed before widespread clinical adoption.
- Modified language throughout: We have systematically replaced definitive statements with more appropriate tentative language:
- 'can contribute' → 'may contribute to'
- 'can enhance' → 'may potentially enhance'
- 'show promise' → 'show preliminary promise in potentially'
- 'represents a novel approach' → 'represents a promising but largely experimental approach'
- Added clinical validation caveats: We have included explicit acknowledgments of the need for larger clinical trials and the current limitations in clinical validation, particularly for sleep spindle modulation therapies.
- Emphasized uncertainties: We have maintained the existing cautionary language regarding orexin system modulation while strengthening the acknowledgment that most interventions require substantial additional research.
These revisions ensure that Section 6 appropriately distinguishes between established evidence and promising but preliminary findings, while maintaining the scientific rigor expected for a review article. The section now clearly communicates both the potential of these approaches and their current limitations, providing readers with a balanced perspective on the state of chronobiological therapeutic innovations in epilepsy.
Reviewer 2 Report
Comments and Suggestions for Authors
RE: Cells-3798413
In the reference manuscript, Kwon et collaborators provide an extensive review of the complex interactions between sleep and epilepsy, along with therapeutic implications of these relationships. The authors emphasize the importance of thalamocortical networks in mediating both spindle sleep activity—a hallmark of NREM sleep—and the occurrence of epileptic spike activity, which can indicate the onset of seizures, particularly absence seizures. The review also explores molecular clock mechanisms essential to epilepsy, illustrating how disruptions in circadian rhythms and alterations in neurotransmitters can increase seizure susceptibility and affect sleep quality. The authors advocate for innovative treatment strategies that prioritize not only effective seizure management but also the enhancement of sleep architecture. Despite the manuscript's clear presentation and engagement with a subject of critical relevance to both researchers and clinicians, there is a notable weakness that somewhat detracts from the overall impact of the review.
Major
A significant oversight is the lack of discussion about the interplay between the thalamus and corticothalamic system, both of which are fundamentally interconnected with the pathophysiology of absence seizures and the emergence of spindles. Including this connection would enrich the review, providing readers with a more nuanced understanding of the underlying mechanisms.
Author Response
We sincerely thank you for their valuable feedback regarding the interplay between the thalamus and corticothalamic system. We agree that this represents a fundamental aspect of both sleep spindle generation and epileptic pathophysiology that deserved more comprehensive discussion.
Major Revision Addressing the Thalamus-Corticothalamic System Interplay:
We have significantly enhanced the manuscript to address this important oversight by adding substantial discussion of corticothalamic interactions in multiple sections:
- Section 2.1 (Sleep Spindles as Guardians of Memory Consolidation): We added a new paragraph that specifically discusses the bidirectional nature of thalamocortical circuits and the critical role of corticothalamic feedback loops. This addition explains how layer VI pyramidal neurons provide essential feedback that modulates thalamic oscillations and maintains coherent spindle activity, creating the reverberating loops fundamental to both normal sleep architecture and epileptic dysfunction [15,16,21,23].
- Section 2.2 (Epileptic Spikes as Circuit Hijackers): We added a substantial new paragraph that directly addresses the mechanistic relationship between sleep spindles and absence seizures within the context of corticothalamic dysfunction. This addition explains how absence seizures represent a pathological hijacking of the same corticothalamic networks that generate sleep spindles, both relying on synchronized oscillations between thalamic relay neurons, TRN cells, and cortical pyramidal neurons [118-122]. The enhanced discussion clarifies how these shared circuits explain why children with absence epilepsy exhibit altered sleep spindle characteristics and establishes corticothalamic dysfunction as a common pathway in both sleep disorders and epileptic encephalopathies.
- Enhanced K-complex discussion: We expanded the existing content to demonstrate how K-complexes are generated through corticothalamic feedback loops that disrupt thalamic spindling [112], illustrating the complex interplay within the same circuits that generate sleep spindles.
- Mechanistic framework: The revised sections now clearly establish that the strength and timing of corticothalamic communication determines whether networks generate protective sleep spindles or pathological epileptic discharges, directly addressing the reviewer's concern about the interconnection between these systems.
- Integration with existing content: The new corticothalamic discussions seamlessly integrate with our existing coverage of memory consolidation mechanisms, neurobiological machinery, and hippocampal coupling, providing a comprehensive view of thalamocortical circuit function in both health and disease.
- Updated references: We have added five new high-quality references [118-122] that specifically support the absence seizure-spindle relationship and corticothalamic mechanisms, including recent systematic reviews and original research from leading epilepsy journals.
These revisions provide readers with a more nuanced understanding of how corticothalamic circuits serve as both the substrate for normal sleep oscillations and the vulnerable networks that become hijacked in epilepsy. The enhanced discussion better positions our arguments for therapeutic interventions that target these bidirectional circuit interactions rather than focusing solely on seizure suppression.
We believe these revisions significantly strengthen the manuscript and comprehensively address the reviewer's concerns about the fundamental interconnections between thalamic and corticothalamic systems in the pathophysiology of both sleep disorders and epilepsy.
Round 2
Reviewer 1 Report
Comments and Suggestions for Authors
Authors have addressed my previous concerns and I have no further comments.
Reviewer 2 Report
Comments and Suggestions for Authors
The authors have responded to my comments, offering detailed insights that reflect their commitment to enhancing the quality of their work.